# Multiscale Joint Optimization Strategy for Retinal Vascular Segmentation

**DOI:** 10.3390/s22031258

**Published:** 2022-02-07

**Authors:** Minghan Yan, Jian Zhou, Cong Luo, Tingfa Xu, Xiaoxue Xing

**Affiliations:** 1College of Electronic Information Engineering, Changchun University, Changchun 130012, China; 200401085@mails.ccu.edu.cn (M.Y.); 200401084@mails.ccu.edu.cn (J.Z.); 190401069@mails.ccu.edu.cn (C.L.); 2School of Optics and Photonics, Beijing Institute of Technology, Beijing 100081, China; ciom_xtf1@bit.edu.cn

**Keywords:** medical image, retinal vascular, matching filter, multiscale, particle swarm optimization algorithm

## Abstract

The accurate segmentation of retinal vascular is of great significance for the diagnosis of diseases such as diabetes, hypertension, microaneurysms and arteriosclerosis. In order to segment more deep and small blood vessels and provide more information to doctors, a multi-scale joint optimization strategy for retinal vascular segmentation is presented in this paper. Firstly, the Multi-Scale Retinex (MSR) algorithm is used to improve the uneven illumination of fundus images. Then, the multi-scale Gaussian matched filtering method is used to enhance the contrast of the retinal images. Optimized by the Particle Swarm Optimization (PSO) algorithm, Otsu algorithm (OTSU) multi-threshold segmentation is utilized to segment the retinal image extracted by the multi-scale matched filtering method. Finally, the image is post-processed, including binarization, morphological operation and edge-contour removal. The test experiments are implemented on the DRIVE and STARE datasets to evaluate the effectiveness and practicability of the proposed method. Compared with other existing methods, it can be concluded that the proposed method can segment more small blood vessels while ensuring the integrity of vascular structure and has a higher performance. The proposed method has more obvious targets, a higher contrast, more plentiful detailed information, and local features. The qualitative and quantitative analysis results show that the presented method is superior to the other advanced methods.

## 1. Introduction

Retinal vascular image segmentation is an important topic in medical image research, which can effectively assist doctors in the clinical diagnosis and treatment of rapid cardiovascular diseases, diabetes and other diseases. In recent years, many scholars have studied retinal vascular image segmentation and achieved some results. However, due to the complexity of retinal images and the influence of noise and light factors in the image acquisition process, accurate retinal vascular image segmentation is still a challenging task [1,2,3]. Two-dimensional color fundus images and 3D Optical Coherence Tomography (OCT) images are the commonly used images for ophthalmic diseases. OCT technology can provide high-resolution retinal images. However, OCT is expensive, images are difficult to acquire, and images need to be registered for vessel segmentation. Color fundus copy is a non-invasive and painless image of the inner wall of the eye taken at different angles using a fundus camera. More importantly, it allows direct visualization of retinal vascular lesions and other lesions such as microaneurysms, hemorrhages, neovascularization, hard exudates, and absorbent cotton spots. Therefore, we choose color fundus images for retinal vessels segmentation studies [4,5,6].

At present, retinal vascular segmentation methods are mainly divided into supervised learning and unsupervised methods. Among the supervised segmentation methods, deep learning-based algorithms are currently a hot research topic. Reference [7] proposed a retinal vessel segmentation method based on cross-channel learning. This method remolds the task of segmentation as a problem of cross-modality data transformation from retinal image to vessel map. Instead of a single label of the center pixel, the network can output the label map of all pixels for a given image patch. The reported accuracy, sensitivity and specificity of their method are 0.9527, 0.7569 and 0.9816, respectively, on the DRIVE database and 0.9628, 0.7726 and 0.9844 for the STARE database. Reference [8] proposed a deep learning method based on segment-level loss which emphasizes more on the thickness consistency of thin vessels in the training process. The average accuracy, sensitivity and specificity are 0.9542, 0.7653 and 0.9818 for DRIVE and 0.9612, 0.7581 and 0.9846 for STARE. In order to make a trade-off between the number of identified overall vascular structures and how to accurately segment a single vessel, [9] proposed a dynamic depth network for retinal vascular segmentation. The average accuracy is 0.9780 for STARE. The method proposed in reference [10] is composed of a convolution neural network based on a simplified version of u-net architecture. The network receives small blocks extracted from the original image as input and uses a new loss function considering the distance between each pixel and the vascular tree for training. The sensitivity, specificity and accuracy mean values are 0.8597, 0.9690 and 0.9563, respectively, on the DRIVE database and 0.8441, 0.9764 and 0.9635 for the STARE database.

Unsupervised segmentation methods can be divided into filter-based methods, clustering based methods, tracking based methods, morphology-based methods and threshold-based methods. Tolias et al. [11] first proposed automatic retinal vascular segmentation method. Chaudhuri et al. [12] proposed a two-dimensional matching filtering method. Odstrcilik et al. [13] proposed an improved matching filtering method. Gabor [14] first proposed Gabor filter theory, then Daugman [15] designed a two-dimensional Gabor filter based on this theory. Remco [16] applied Cake filter in retinal vascular segmentation. Reference [17] presented a multi-scale Frangi filter-based algorithm. Wang et al. [18] proposed an improved morphological and OTSU retinal vascular segmentation method. In addition to these methods, the blood vessel segmentation method based on fuzzy C-means clustering (FCM) [19] and K-means [20] are also popular. Reference [21] proposed a method based on vascular tracking to extract the retinal vascular part. Fraz et al. [22] proposed a segmentation method based on morpho-logical top-hat transform. Reference [23] proposed a verification-based multi-threshold detection adaptive local threshold method.

Most supervised segmentation methods utilize manually designed features to model the retinal vasculature. However, manual design of features is a heuristic and laborious process that relies heavily on experience and skills. In addition, the parameters used in the algorithm usually need to be carefully tuned in order to address pathology, image noise, and other complex situations. The unsupervised segmentation methods have a fast operation time and low cost, and can obtain better segmentation performance. But these methods are susceptible to noise and cannot capture small blood vessels which provide important information for the detection of diseases. 

In this paper, a new unsupervised method is presented. We design a multi-scale joint optimization strategy for retinal vascular segmentation, which can segment more small blood vessels and improve the segmentation accuracy. The segmentation results are beneficial for the diagnosis, screening and treatment of cardiovascular and ophthalmologic diseases. The main contributions of this paper are as follows: (1) the MSR algorithm is used to adjust the brightness of the image and reduce noise. (2) A multi-scale Gaussian matched filtering method is proposed to enhance the contrast of the images. (3) The PSO algorithm is used to optimize OTSU three threshold, accelerate the speed and improve the accuracy.

The remainder of this paper is arranged as follows. In Section 2, the proposed method framework is given, the relevant theories used in each step of the proposed method are explained and the results of each step are shown. Section 3 shows the experimental results and analysis. Section 4 summarizes the paper.

## 2. Proposed Methodology

### 2.1. Overview

The flow chart of the algorithm for blood vessel segmentation in this paper is shown in Figure 1. As shown in Figure 1, a retinal segmentation method based on multi-scale joint optimization strategy is mainly divided into four stages: image pre-processing, vascular feature extraction, image multi-threshold segmentation, and image post-processing. Firstly, we use MSR to adjust the brightness of the image and reduce noise, and the green channel is extracted as the original image for subsequent processing. Secondly, the Multi-scale Gaussian matched filtering method is proposed to enhance the contrast of the images and extract the features of the blood. Then, PSO is used to optimize OTSU three thresholds for image segmentation. Finally, the binarized image is processed by breakpoints connection, denoising and edge contour removal. To illustrate the various steps of the algorithm in detail, Figure 2 shows the amplification images of all the output images in important steps.

### 2.2. Image Pre-Processing

#### MSR Algorithm

The basic idea of Retinex [24] is that the object color perceived by the human visual system is determined by the reflection properties of the object surface and which has a slight relationship with the incident light information. Assuming the original image is $S\;\left(x,\;y\right)$, then,
(1)S(x,y)=i(x,y)∗R(x,y) 
where

$\left(x,y\right)\;$represents the coordinate point in two-dimensional space.

$S\left(x,y\right)\;$represents the original image. 

$i\left(x,y\right)\;$represents the illumination image.

$R\left(x,y\right)\;$represents the reflectivity image.

When i is removed from S, the remaining R is an image that eliminates the impact of light, as human visual systems perceive.

As shown in Figure 1, the low contrast between the target blood vessels and the background is not beneficial for the later segmentation. Thus, we use the MSR algorithm to adjust the brightness and enhance the contrast of the retinal image. The MSR algorithm is proposed by Jobson D J et al. [25], and is defined as shown in Equation (2).
(2)$$R_{MSR_{i}}=\sum_{n=1}^{N}w_{n}R_{n_{i}}=\sum_{n=1}^{N}w_{n}(Log[S_{i}(x,y)]-Log[S_{i}(x,y)*F_{n}(x,y)])$$
where

$S_{i}\left(x,y\right)\;$is the ith channel component of the original retinal color image.

$R_{MSR_{i}}$ is the reflection component of the channel.

$R_{n_{i}}$is the incident component (illumination component) of the ith channel at the nth scale.

$F_{n}\;$is the nth Gaussian function. N is the number of scales (to ensure that the MSR al-gorithm has the advantages of both high and low scales, the value of N is generally selected as 3).

$\omega_{n}\;$is the weight of Gaussian convolution at the nth scale.

The red-green-blue three-channel extraction experiment is carried out on the images processed by MSR, and it can be found that the contrast between the target and the background of the red channel is low and the noise of the blue channel is large. The green channel image has a balanced brightness, high contrast and uniform gray distribution. Therefore, the green channel image is selected for subsequent processing.

### 2.3. Vascular Feature Extraction

In this paper, we use Gaussian matching filter to extract the features of the retinal blood vessel. The Gaussian matching filter was first proposed by Chaudhuri et al. [12]. The Gaussian kernel function used in [12] was described as below.
(3)$$K(x,y)=-\exp(-x^{2}/2\sigma^{2}),\left|y\right|<L/2$$
where L is the length of the Gaussian kernel, which indicates the length of blood vessels that can be detected by the filter, we set L to 9, σ is the scale of Gaussian kernel, which represents the vascular cross-sectional extension area that can be detected by the filter. For the vessels at different orientations, the Gaussian kernel should be rotated accordingly. The Gaussian kernel rotates once every 15 degrees from 0 to 180 degrees (θ = 0, 15,..., 180.), and a total of 12 directions are constructed to retain the maximum filtering response of each pixel. The rotation matrix is given by
(4)$$r_{i}=\left[\begin{array}{cc} \cos\theta_{i} & -\sin\theta_{i}\\ \sin\theta_{i} & \cos\theta_{i} \end{array}\right],0\leq \theta \leq \pi$$

Suppose $p=\left[x,y\right]$ is a discrete point in the kernel function, $\theta_{i}\;\left(0\leq \theta_{i}\leq \pi \right)$ is the angle of the $i_{\mathrm{th}}$ kernel function, then the coordinate value of p after rotation is $\bar{p_{i}}=\left(u,v\right)={\mathrm{pr}}_{i}^{T}$, then the $i_{\mathrm{th}}$ template kernel function is
(5)$$K_{i}(x,y)=-exp(-u^{2}/2\sigma^{2}),\forall {\bar{p}}_{i}\in Z$$
where Z is the template field and the value range is:$\;Z=\{(u,\;v),\left|u\right|\leq 3\sigma ,\left|v\right|\leq L/2$. When the vessel length is less than the filter length, the vessel segment is approximately regarded as a straight line. If the vessel width matches the scale of the Gaussian kernel, the output value of the filter is maximum. 

The filtered image is obtained by convolution of the input image with the two-dimensional Gaussian kernel. The mathematical expressions are as follows:(6)$$G(x,y)=K(x,y)*R_{MSRG}(x,y)$$
where *G* represents the filtered image and R represents the input image.

#### 2.3.1. Multi-Scale Matching Filtering

Because the length, width, branch and angle of retinal blood vessels are different, it is difficult to accurately extract the vascular feature information in a single scale. Therefore, this paper selects a multi-scale matching filter to extract the characteristics of vascular images. When the macroscale image is selected for filtering, the coarsest blood vessel is mainly extracted. When the small scale is selected to filter the image, the smallest blood vessel is extracted. After many experiments, when$\;\sigma_{1}=1.9$, the main contour feature of the blood vessel can be effectively extracted. When$\;\sigma_{3}=0.13$, the details of the blood vessel can be effectively extracted. By adding an intermediate scale to filter the image, the following effects can be achieved: (1) denoising while enhancing the extraction of small vessels. (2) The width of smaller vessels will not be overestimated. (3) There is a reasonable filtering response to the blood vessels. When$\;\sigma_{2}=0.5$, images including main contour features and partial details can be obtained.

The effect of vascular information extraction at scales σ of 1.9, 0.5 and 0.13 is shown in Figure 3. As shown in Figure 3, the most features of the retinal blood vessels are extracted, which are beneficial for the subsequent image processing.

#### 2.3.2. Information Fusion of Vascular Characteristics

Multi-scale matched filtering can obtain most of retinal vascular information features information at different scales. In order to effectively enhance the contrast between the target blood vessel and the background and obtain better retinal vascular images, the results of each scale matched filtering are fused. The fusion calculation is given as follows:(7)$$G=\omega_{1}G_{1}+\omega_{2}G_{2}+\omega_{3}G_{3}$$
where

G is the fused image.

$G_{1}{\;G}_{2}{\;G}_{3}\;$are the vascular feature image at${\;\sigma }_{1}=1.9$ $\sigma_{2}=0.5$ and${\;\sigma }_{3}=0.13$.

ω is the weight of each scale superposition.

The fusion results are shown in Figure 4. Figure 4b shows the extraction result of the large scale at $\sigma_{1}=1.9$. Figure 4d displays the extraction result of the small scale$\;\sigma_{3}=0.13$. Figure 4f shows the three-scale extraction result. Meanwhile, in order to show fully the comparison effects of the single and three-scale extraction methods, we magnify the details of the red region respectively. It can be seen from Figure 4 that the extraction effect of the single scale is often not good.

When we use the large scale of$\;\sigma_{1}=1.9\;$to process the image, a lot of small and some main vessels are lost. The vascular structure of the extraction image is incomplete. When we adopt the small scale of$\;\sigma_{3}=0.13$ to process the image, the extraction result has vascular ruptures, poor vascular connectivity and strong noise. Compared with the single-scale filtering, the multi-scale matched filtering method can preserve vascular integrity, effectively extract more small vessels, and reduce the effects of noise.

### 2.4. Image Segmentation

#### 2.4.1. OTSU Algorithm

The OTSU algorithm was first proposed in 1979 [26]; it selects the optimal threshold by maximizing the class variance of the segmented class. The pixels of a given image have L gray levels$\;\left[1,2\ldots \;L\right]$. The number of pixels in level i is $n_{i}$, and the total number of pixels is $N=n_{1}+n_{2}+\ldots n_{L}$.The probability of pixels with gray value i is denoted as $p_{i}$:(8)$$p_{i}=\frac{n_{i}}{N},p_{i}\geq 0,\sum_{i=0}^{L-1}p_{i}=1$$

The given image is divided into $C_{0}$ and $C_{1}$ regions by the threshold t. $C_{0}$ represents the pixel level$\;\left[1,\;\ldots ,\;k\right]$, $C_{1}$ represents the pixel level $\left[k+1,\;\ldots ,\;L\right]$. The probability and average gray value of the region are given by Equations (9) and (10) respectively, and the total mean level $u_{T}$ of the original image is given by Equation (11).
(9)$$w_{0}=\sum_{i=0}^{t}p_{i}\;\;w_{1}=\sum_{i=t+1}^{L-1}p_{i}$$
(10)$$u_{0}=\sum_{i=0}^{t}ip_{i}/w_{0}\;\;u_{1}=\sum_{i=t+1}^{L-1}ip_{i}/w_{1}$$
(11)$$u_{T}=\sum_{i=0}^{L-1}ip_{i}$$

The following two relationships expressed by Equation (12) can be easily verified:(12)$$w_{0}u_{0}+w_{1}u_{1}=u_{T},w_{0}+w_{1}=1$$

The objective function of the OTSU method can be defined as
(13)$$\sigma^{2}{}_{B}=w_{0}{\left(u_{0}-u_{T}\right)}^{2}+w_{1}{\left(u_{1}-u_{T}\right)}^{2}$$

When$\;\sigma_{B\left(t\right)}^{2}={\mathrm{Argmax}\sigma }_{B}^{2}$, t obtains the optimal value. Extending the OTSU single threshold to multiple thresholds with interclass variances: we can obtain the best threshold combination ($t_{1}$, $t_{2}$, ... $t_{m}$) when the maximum is obtained. The specific calculation can be described as below:(14)$$\sigma^{2}{}_{B}(t_{1}t_{2}\ldots t_{m})=w_{0}{\left(u_{0}-u_{T}\right)}^{2}+w_{1}{\left(u_{1}-u_{T}\right)}^{2}+\ldots w_{m}{\left(u_{m}-u_{T}\right)}^{2}$$

#### 2.4.2. PSO Algorithm

PSO algorithm [27] is a swarm intelligence algorithm proposed by simulating bird swarm foraging, which is used to find the solution that makes the objective function obtain maximum or minimum. In the PSO algorithm, the bird swarm is assumed to be a particle with no mass and volume in N-dimensional space, and each particle i is a candidate solution. Each particle passes through speed and position to find the best in the workspace. Each particle moves around by its own ‘speed’ in the search space, and the speed is the distance travelled by the particle from one position to the current position. Each particle is affected by its individual best realization position pbest and the group global best position gbest (solution of the problem). The initialization of PSO algorithm is a group of random particles, namely random solution. The speed and position of particle i in d-dimensional search space update according to Equations (15) and (16), the specific parameter settings are shown in Table 1.
(15)$$V_{id}^{K}=wV_{id}^{K-1}+c_{1}r_{1}\left({\mathrm{pbest}}_{\mathrm{id}}-x_{id}^{K-1}\right)+c_{2}r_{2}$$
(16)$$x_{id}^{K}=x_{id}^{K-1}+v^{K-1}{}_{id}$$
where

$v_{\mathrm{id}}^{K}\;$is the d-dimensional component of the velocity of i particle in iteration 𝑘.

$x_{\mathrm{id}}^{K}\;$is the d-dimensional component of the position of i particle in iteration 𝑘.

$c_{1}\;$and $c_{2}$ are acceleration constants, which are used to adjust the learning step size.

$r_{1}$and $r_{2}$ are two random functions with the range of values $\left[0,\;1\right]\;$to increase the search randomness.

ω is the inertia weight factor used to adjust the search range of solution space.

#### 2.4.3. OTSU Image Segmentation Based on PSO (OTSU-PSO Algorithm)

The background, target and noise of the pre-processed image are at different gray levels. In order to obtain the best segmentation effects, we use multi-threshold to segment the image, and the image can be divided into multiple regions with a multi-gray level. However, it would take too much time to search an optimal threshold combination in the full gray range. To simplify the calculation and improve the operation speed, we used the PSO algorithm to search the optimal threshold combination. It was found experimentally that when the number of the segmentation threshold combination is 3, a better segmentation effects can be achieved. Since the expert segmentation results in the retinal image data set are all binary images, in order to ensure the accuracy of the evaluation index calculation, it is necessary to use the OTSU single threshold to transform the retinal vascular image obtained after the OTSU-PSO algorithm into a binary image, and then the final result can be obtained after image post-processing (see Section 2.5 for details of post-processing). The final segmentation results of OTSU-PSO algorithm are shown in Figure 5. The specific steps of OTSU-PSO algorithm are shown in Table 2.

### 2.5. Image Post-Processing

The image obtained by the OTSU-PSO algorithm is re-segmented to get the segmented retinal vascular image. The segmented retinal vascular image has the following problems: (1) Some blood vessels are broken; (2) The field edge of fundus camera with false segmentation exists; (3) Noise is also enhanced when detailed features are extracted. In order to solve the problems and compare with the expert segmentation results of retinal image dataset, we post-process the image. The specific operation steps are as follows:(1).the median filter is used to denoise the image and connect the broken blood vessels.(2).the morphological processing is used to connect domain area and remove the large noise.(3).the mask image is extracted from the source retinal image, and the difference image between the source retinal image and the mask image is obtained.(4).The difference image is binarized by the OTSU algorithm, and then the binary image is expanded by the morphological processing.(5).The segmented vascular image is subtracted from the expanded edge image to get the final output image.

Randomly selected images on the Drive dataset are used to test the PSO-based OTSU three-threshold segmentation results. The effects of the multi-threshold and single-threshold segmentation methods are shown in Figure 5. As shown in Figure 5, there are more small blood vessels lost in the single-threshold segmentation image, and the main blood vessels have structural fracture. Compared with the single-threshold segmentation method, a three-threshold segmentation image has more small blood vessels and better connectivity.

## 3. Results and Discussion

### 3.1. Experimental Environment and Datasets

All the experiments are implemented in Matlab2016a (Mathworks, Natick, MA, USA) on 2.30 GHz processor with 3.8 GB RAM. We use two publicly available datasets, DRIVE dataset [2] and STARE dataset [28] to evaluate the performance of the proposed method. The DRIVE dataset contains a total of 40 color retinal images with a resolution of 565 × 585. It is divided into two sets: a testing set and a training set, and each set contains 20 images. The training set includes an artificial split set that is completed by one expert. The testing set includes two manual segmentation sets completed by two experts. The STARE dataset contains 20 color retinal images with the resolution of 700 × 605. The STARE dataset contains two sets of images manually segmented by two experts. There is no separate training and test set available for this dataset.

### 3.2. Segmentation Evaluation Index

In order to better judge the segmentation effect of the model, it is necessary to compare the segmentation results with the ground truths manually marked by experts. Three most common evaluation metrics, Accuracy (Acc), Sensitivity (Se) and Specificity (Sp) are used to evaluate the segmentation results. Acc represents the ratio of the number of correctly segmented pixels to the total pixels. Se represents the ratio of the number of correctly segmented vascular points to total pixels. Sp represents the ratio of the number of correctly segmented background points to the total pixels. The higher the value, the higher the success. The three-evaluation index can be described as
(17)$$Acc=\frac{TP+TN}{TP+FN+TN+FP}$$
(18)$$Se=\frac{TP}{TP+FN}$$
(19)$$Sp=\frac{TN}{TN+FP}$$
where

TP (true-positive) is the number of points correctly segmented into blood vessels.

FP (false-positive) is the number of vascular points that are incorrectly segmented. 

TN (true-negative) is the number of points correctly segmented as background. 

FN (false-negative) represents the number of background points that are wrongly segmented.

The above three measures metrics are based on a subset of the following four basic quantities: TP,TN, FP and FN. The measure methods assume that the pixels are independent of each other. Hence, they may cause dependency flaw. Thus, we also adopt Structure Similarity Measure (SSIM) proposed by Wang [29] and Structural Measure(S-measure) proposed by Deng [30] to evaluate the segmentation results.

### 3.3. Experimental Results and Analysis

The segmentation comparison results on DRIVE and STARE datasets are shown in Figure 6 and Figure 7 and Table 3 and Table 4.

Figure 6a and Figure 7a are the original images of the DRIVE and STARE datasets. Figure 6b and Figure 7b are the segmentation results of the proposed method, which are displayed in red. Figure 6c and Figure 7c are the results made by the first expert, which are displayed in green Figure 6d and Figure 7d are the results made by the second expert, which are displayed in green. Figure 6e,f and Figure 7e,f are the differences between the proposed method and the segmentation results of the first expert and the second expert. According to the color scheme of Figure 6 and Figure 7, the yellow represents the vascular pixels that are correctly segmented. As demonstrated in Figure 6 and Figure 7, the segmentation results on DRIVE and STARE datasets show that there are some red parts, indicating that the proposed method can segment more small blood vessels while ensuring the integrity of the main blood vessels.

In order to analyze the effectiveness of the method adopted in this paper, the quantitative results of this experiment on DRIVE dataset, STARE dataset are shown in Table 3. As shown in Table 3, we use the segmentation results of the first expert as the gold standard to calculate the evaluation index of DRIVE dataset, the average specificity, sensitivity and accuracy of the method are 0.9702, 0.7577 and 0.9514; we also use the segmentation results of the first expert as the gold standard to calculate the evaluation index of STARE dataset, the average specificity, sensitivity and accuracy of the method are 0.9699, 0.7763 and 0.9579.

In order to overcome the dependency flaw of the above three measure parameters, we also use the SSIM and S-measure to evaluate the effectiveness of the proposed method. The higher the values, the better performance. The performance of the proposed method on the two public datasets are listed in Table 4. We calculate the SSIM and S-measure between the segmentation results obtained by the proposed algorithm and the two ground-truth images from the two experts. In addition, for comparison, we compute the SSIM and S-measure between the two experts’ segmentation results. It can be concluded that the proposed method can achieve higher values of SSIM and S-measure, and the segmentation results are better.

### 3.4. Comparison with Other Methods

In order to intuitively compare the segmentation performance of retinal vessels, the results of this experiment are compared with those of the three methods of the two-dimensional matching filter (M1), the linear tracking morphological (M2) and cap transformation (M3), as shown in Figure 8. In Figure 8, the images in the first and second rows are randomly selected in the Drive test set, and the images in the third and fourth rows are randomly selected in the STARE dataset. The comparison results show that our method is superior to other three methods, which can segment more small vessels while maintaining structural integrity. The segmentation results are comparable to expert manual segmentation results, which is beneficial for the disease diagnosis.

Table 5 gives the comparison results of the proposed method with those of state-of-the-art methods for the two datasets. The comparison methods include five supervised based methods and five unsupervised based algorithms. Moreover, the results of the ten methods are from their paper. The value in bold represents the performance of the proposed method.

Compared with the unsupervised based methods, for DRIVE dataset, the Sp of the presented method is 0.0224 lower than maximum value, the Acc of the presented method is 0.0031 lower than maximum value, but the presented method can obtain the highest Se; for STARE dataset our method achieves the highest Se and Acc, while the Sp is 0.0116 lower than maximum value. Compared with the results of the supervised based methods, for the DRIVE dataset, the Sp of the presented method is 0.0114 lower than the maximum value, the Acc of the presented method is 0.0048 lower than maximum value, the Se of the presented method is 0.0076 lower than maximum value; for the STARE dataset, the Se is the maximum, while the Acc and Sp are 0.0049 and 0.0147 lower than the maximum, respectively. In general, when jointly regarding the performance measures of Se, Sp and Acc, our approach outperforms state-of-the-art methods on the DRIVE and STARE datasets. Our method has less calculation, higher accuracy and a certain robustness.

## 4. Conclusions

In this paper, we present a multiscale joint optimization strategy for retinal vascular segmentation. The use of the multi-scale matching filtering method can enhance the contrast between the target blood vessels and the background. The optimization strategy utilized PSO can get the optimal segmentation threshold combination. In order to evaluate the effectiveness and applicability of the proposed method, the experiments are implemented on the DRIVE and STARE datasets. The qualitive and quantitative analysis demonstrates that the proposed method outperforms other existed methods and has strong robustness. The segmented images of the presented method have more small blood vessels and better integrity of vascular structure, which is beneficial for the diagnosis of diseases. The main purpose of retinal vessel segmentation proposed in this paper is to assist doctors in the diagnosis of cardiovascular and cerebrovascular diseases. In the future, we plan to classify fundus related diseases based on the segmented retinal vessels, such as glaucoma, senile macular edema and so on. Limited by the currently available datasets, in the future, we will build a new dataset that contains retinal images from patients of diabetic retinopathy, glaucoma and other ophthalmic diseases, and the new dataset can be used to evaluate the capability of the algorithms in handling pathological images.

## Figures and Tables

**Figure 1 sensors-22-01258-f001:**
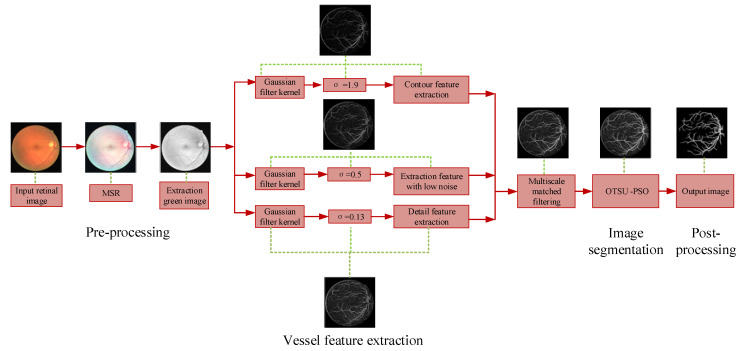
The flow chart of the proposed algorithm for vessel extraction.

**Figure 2 sensors-22-01258-f002:**
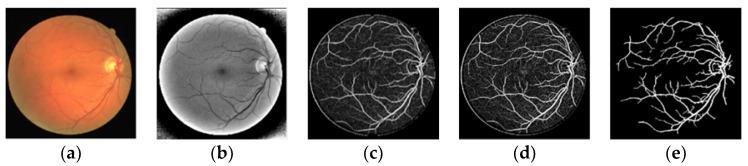
Multiscale joint Optimization strategy. (**a**) Original color image; (**b**) Pre-processing result; (**c**) Multi-scale filtering; (**d**) OTSU image segmentation based on PSO (OTSU-PSO); (**e**) Post-processing.

**Figure 3 sensors-22-01258-f003:**
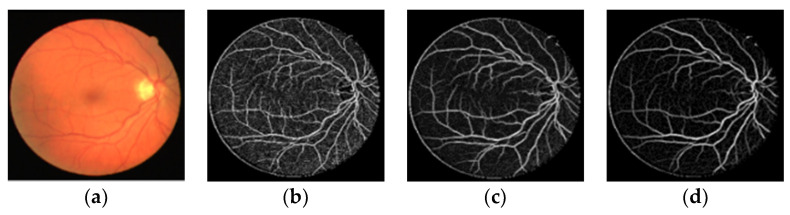
Retinal vessels information extraction image. (**a**) Original image; (**b**) $\sigma_{1}=1.9;$ (**c**)$\;\sigma_{2}=0.5;\;$(**d**) $\sigma_{3}=0.13$.

**Figure 4 sensors-22-01258-f004:**
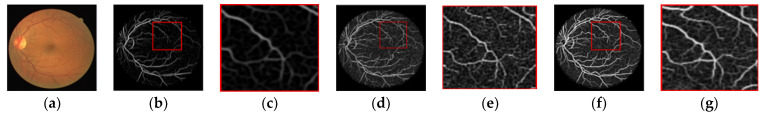
Comparison of multi-scale and single-scale vascular extraction. (**a**) original image; (**b**) $\sigma_{1}=1.9;$ (**c**) Detail amplification of $\sigma_{1}=1.9;$ (**d**) $\sigma_{3}=0.13;$ (**e**) Detail amplification of ${\;\sigma }_{3}=0.13;$ (**f**) multi-scale fusion vascular extraction image; (**g**) Detail amplification of multi-scale.

**Figure 5 sensors-22-01258-f005:**
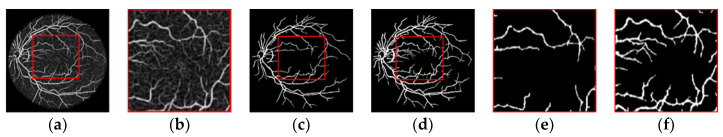
Comparison of multi-threshold and single-threshold segmentation results. (**a**) Multi-scale matched filtering figure; (**b**) Detail amplification of Multi-scale matched filtering; (**c**) Single-threshold segmentation figure; (**d**) Three-threshold segmentation figure; (**e**) Detail amplification of Single-threshold segmentation; (**f**) Detail amplification of Three-threshold segmentation.

**Figure 6 sensors-22-01258-f006:**
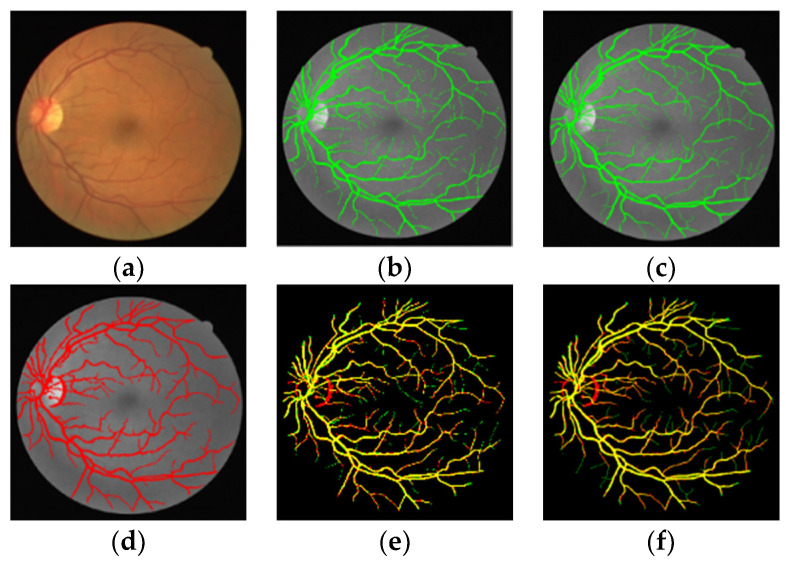
Comparison results of the proposed algorithm with that of experts on DRIVE 01_test. (**a**) DRIVE image; (**b**) Expert 1 segmentation result; (**c**) Expert 2 segmentation result; (**d**) Segmentation results of the proposed method; (**e**) Difference between segmentation result of proposed method and expert 1 segmentation result; (**f**) Difference between segmentation result of proposed method and expert 2 segmentation result.

**Figure 7 sensors-22-01258-f007:**
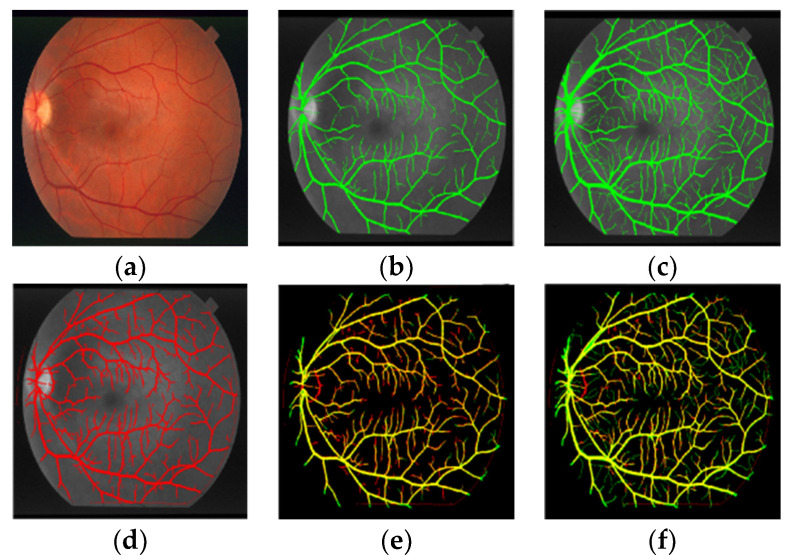
Comparison results of the proposed algorithm with that of experts on STARE im0255. (**a**) STARE image; (**b**) Expert 1 segmentation result; (**c**) Expert 2 segmentation result; (**d**) Segmentation results of the proposed method; (**e**) Difference between segmentation result of proposed method and expert 1 segmentation result; (**f**) Difference between segmentation result of proposed method and expert 2 segmentation result.

**Figure 8 sensors-22-01258-f008:**
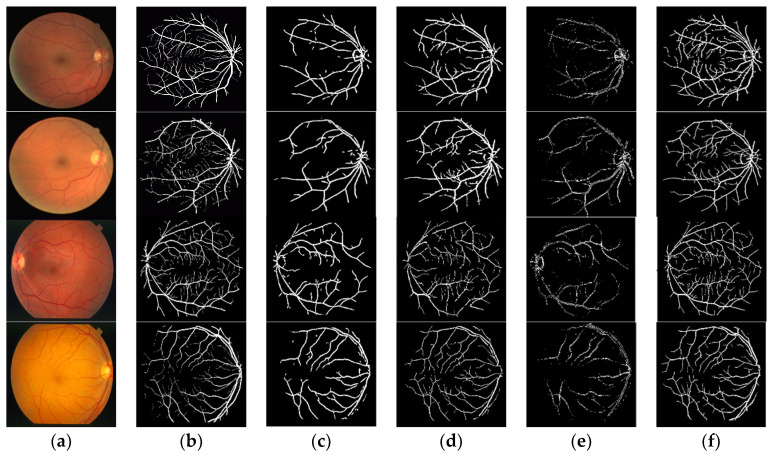
Blood vessel segmentation in retinal images using the proposed method. (**a**) Original image; (**b**) The second expert; (**c**) M1; (**d**) M2; (**e**) M3; (**f**) Proposed algorithm.

**Table 1 sensors-22-01258-t001:** PSO Parameter Settings.

Parameter Value	PSO
Population size (N)	40
Inertia weight (w)	0.5
Learning constants	$c_{1}=c_{2}=2$
Max.Iteration (M)	20
$\mathrm{Initial}\;\mathrm{Pulse}\;\mathrm{rate}\;(r_{1},r_{2}$)	X
X: Not parameter value	

**Table 2 sensors-22-01258-t002:** OTSU-PSO algorithm.

**Input:** number of iterations M, population size N, dimension D. **Output:** the optimal threshold combination (gbest _ position (i), i is the threshold number).
Step 1: Initialize the velocity and position of particles, individual extremum pbesti and global extremum gbest. Step 2: Equation (14) is used to calculate the fitness value of each particle to update the individual extremum pbesti and the global extremum gbest. Step 3: Update the particle velocity and position of the particle according to the Equations (15)–(16). Step 4: Determine if the iteration stop condition is satisfied, then the algorithm ends. Otherwise turn to Step 2, continue to iterative cycle, and finally find the optimal solution.

**Table 3 sensors-22-01258-t003:** Test results of the proposed algorithm on DRIVE dataset and STARE dataset.

DRIVE	Acc	Se	Sp	STARE	Acc	Se	Sp
01_test	0.9404	0.8938	0.9450	im0001	0.9424	0.6506	0.9656
02_test	0.9567	0.7809	0.9768	im0002	0.9387	0.5427	0.9650
03_test	0.9401	0.7380	0.9625	im0003	0.9572	0.6199	0.9829
04_test	0.9535	0.7419	0.9749	im0004	0.9426	0.8315	0.9455
05_test	0.9545	0.7281	0.9779	im0005	0.9475	0.7248	0.9680
06_test	0.9445	0.7010	0.9707	im0044	0.9681	0.7815	0.9681
07_test	0.9512	0.7132	0.9751	im0077	0.9596	0.7697	0.9748
08_test	0.9500	0.7047	0.9730	im0081	0.9549	0.6970	0.9757
09_test	0.9559	0.7082	0.9777	im0082	0.9696	0.8445	0.9790
10_test	0.9560	0.7675	0.9729	im0139	0.9541	0.7254	0.9731
11_test	0.9487	0.7548	0.9678	im0162	0.9669	0.7477	0.9852
12_test	0.9553	0.7551	0.9742	im0163	0.9745	0.8564	0.9838
13_test	0.9537	0.6764	0.9837	im0235	0.9660	0.8739	0.9733
14_test	0.9527	0.7947	0.9666	im0236	0.9677	0.8957	0.9734
15_test	0.9067	0.8514	0.9110	im0239	0.9571	0.9034	0.9601
16_test	0.9617	0.7334	0.9844	im0240	0.9323	0.9248	0.9326
17_test	0.9597	0.6609	0.9872	im0255	0.9675	0.8120	0.9832
18_test	0.9655	0.7673	0.9826	im0291	0.9706	0.7900	0.9775
19_test	0.9580	0.8813	0.9650	im0319	0.9707	0.7520	0.9769
20_test	0.9628	0.8016	0.9756	im0324	0.9496	0.7910	0.9542
Mean	0.9514	0.7577	0.9702	0.9579	0.9729	0.7762	0.9699

**Table 4 sensors-22-01258-t004:** Performance Metrics of Proposed Method.

Datasets	Contrast	SSIM	S-Measure
DRIVE	Proposed vs. Expert 1	0.7385	0.7982
	Proposed vs. Expert 2	0.6220	0.8069
	Expert1 vs. Expert2	0.5142	0.8366
STARE	Proposed vs. Expert 1	0.7636	0.7986
	Proposed vs. Expert 2	0.7207	0.7708
	Expert1 vs. Expert2	0.7122	0.7793

**Table 5 sensors-22-01258-t005:** Comparison of proposed method and other methods on DRIVE and STARE datasets.

Methods		Years	Acc	Se	Sp	Acc	Se	Sp
			DRIVE				STARE	
Supervised	Li et al. [7]	2016	0.9527	0.7569	0.9816	0.9628	0.7726	0.9844
	Dasgupta et al. [31]	2016	0.9533	0.7569	0.9792	-	-	-
	Yan et al. [8]	2018	0.9542	0.7653	0.9801	0.9612	0.7581	0.9846
	Yang et al. [32]	2019	0.9421	0.7560	0.9696	0.9477	0.7202	0.9733
	Adapa et al. [33]	2020	0.9450	0.6994	0.9811	0.9486	0.6298	0.9839
Unsupervised	Biswal et al. [34]	2018	0.9545	0.7100	0.9700	0.9495	0.7000	0.9700
	Ben et al. [35]	2018	0.9389	0.6887	0.9765	0.9388	0.6801	0.9711
	Wang et al. [18]	2019	0.9382	0.5686	0.9926	0.9460	0.6378	0.9815
	Roy et al. [36]	2019	0.9295	0.4392	0.9622	0.9488	0.4317	0.9718
	YUAN et al. [17]	2020	0.9500	0.7100	0.9700	-	-	-
Proposed		2021	0.9572	0.7798	0.9758	0.9579	0.7762	0.9699

## Data Availability

Not applicable.

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
