# Peer review of "Multiscale Joint Optimization Strategy for Retinal Vascular Segmentation"

_sensors, 2022, doi:10.3390/s22031258_

Round 1

Reviewer 1 Report

In this paper the authors propose a multi-scale joint optimization strategy for retinal vascular segmentation. The authors use MSR to to adjust the brightness of the image and reduce noise,multi-scale Gaussian matched filtering method to enhance the contrast of the images, and the Optimized by particle swarm optimization (PSO) algorithm to optimize OTSU three thresholds for image segmentation. 

After all this process, and a postprocessing step, the segmentation results obtained are promising to assist doctors in the clinical diagnosis of some diseases such as cardiovascular diseases, diabetes and others.

In recent years, much progress has been made on these topics using deep learning. It would be necessary to update the state of the art with current work in order to really see if the proposed method improves the current state of the art. An example of a newer paper about this topic can be found here https://doi.org/10.1016/j.cmpb.2021.106081. with an extensive comparison with other methods for the exact two datasets evaluated.

At the end of the introduction it would be good to add a paragraph with the structure that the article will have and what will be developed in each section

Authors should include the meaning of acronyms in the first appearance (MSR, KNN , CNN and so on)

Some future works should be added to the conclusions section.

Reviewer 2 Report

The paper is devoted to an important problem related to processing of retinal images. Several specific comments are bellow:

Major comments:

  1. Section 1 INTRODUCTION: I suggest to specify more precisely the goal of the paper. Is the registration of images of the same individuals and their time evolution studied as well? Associated references including registration and deep learning methodology should be added, for instance:

[1] Procházka A., at al:  Registration and Analysis of Retinal Images for Diagnosis and Treatment Monitoring. Proc. of the Int. Workshop on Computational Intelligence for Multimedia Understanding, Paris, pp T5/1-T5/4, 2014

[2] Khanal A., at al:  Dynamic Deep Networks for Retinal Vessel Segmentation, Frontiers in Computer Science, 2:35, 2020

  1. Section 2 PROPOSED METHODOLOGY: The proposed method should be better and more clearly described. The use of the two dimensional Gaussian kernel and the selection of its parameters should be more clearly presented.

  1. Section 3 RESULTS AND DISCUSSION: Sets of images should be better specified and results achieved better described. Some mistakes should be corrected (page 10, line 329: “Fig. 6(d) and Fig 6. (d)” >>  “Fig. 6(d) and Fig 7(d)”, …  

  1. Section 3.4 COMPARISON WITH OTHER METHODS: Results in Table 5 should be better described. Were these results evaluated by authors for the same sets of images? Publication years (the second column) does not correspond with references in some cases.

  1. Section 4 CONCLUSION: Furter research should be mentioned here

Minor comments:

  1. The whole text should be carefully examined and corrected. The word “where” after Eqs. (1), (2), (4), (7), (16), (19),… should be without intend: Where >> where; the number of Eq. (2) should be on the same line as the equation,…

  1. Language mistakes should be corrected and the formal level of the whole text increased (spaces in captions added, the flowchart in Table 2 better structured; Tables 2, 3 more compact, …)

  1. All variables should be in italics, matrices in bold

  1. References should be examined to include all facts and to be consistent ( [1], [13], [14], [17], [20], [21], [22], [23], [30], … )

Reviewer 3 Report

The authors present a segmentation methodology for eye fundus images. The topic is of high interest in today's medical field. The authors propose a ML segmantation approach, combining filtering, and optimisation approaches for reconstructing the arterial tree.

A basic problem with such approaches, is that the evaluation metrics are based on a pixel-to-pixel comaprison, and not on the finding of a specific pattern comprised of a number of pixels such as e.g. a blood vessel.

Of course if an algorithm can classify accurately all the pixels, then yes, we would expect to find the patterns we are interested in without a problem. In my view this problem can be encountered if the authors define a 2-D moving window which can define ROIs in the image and take into account the probability of pixels achieving the highest brightness in the gray scale (or the two brightest scales) and combine it then with the outcomes of the algorithm.

No matter what, I would suggest that the authors address this issue in their submission when revising.

Further, it seems to me that in a numer of equations the variable letters do not have a consistent size, and in some cases this creates confusion to the reader. Please make sure that for example the velocity parameter looks the same in all spots it appears.

Round 2

Reviewer 1 Report

Thanks for all the changes. The paper is now better and includes a deeper comparison with other methods. I think it is ready for publishing 

Reviewer 2 Report

All comments were answered.

Reviewer 3 Report

Thank you for revising your paper appropriately.